# Effects of Different Types of LAB on Dynamic Fermentation Quality and Microbial Community of Native Grass Silage during Anaerobic Fermentation and Aerobic Exposure

**DOI:** 10.3390/microorganisms11020513

**Published:** 2023-02-17

**Authors:** Jiawei Zhang, Yichao Liu, Zhijun Wang, Jian Bao, Muqier Zhao, Qiang Si, Pengbo Sun, Gentu Ge, Yushan Jia

**Affiliations:** 1Key Laboratory of Forage Cultivation, Processing and High Efficient Utilization of Ministry of Agriculture and Rural Affairs, Inner Mongolia Agricultural University, Hohhot 010019, China; 2Key Laboratory of Grassland Resources, Ministry of Education, Inner Mongolia Agricultural University, Hohhot 010019, China; 3College of Grassland, Resources and Environment, Inner Mongolia Agricultural University, Hohhot 010019, China

**Keywords:** native grass, lactic acid bacteria, aerobic stability, microbial community, silage

## Abstract

Silage of native grasses can alleviate seasonal forage supply imbalance in pastures and provide additional sources to meet forage demand. The study aimed to investigate the effects of *Lactobacillus plantarum* (LP), *Lactobacillus buchneri* (LB), and *Lactobacillus plantarum* in combination with *Lactobacillus buchneri* (PB) on the nutritional quality, fermentation quality, and microbial community of native grass silage at 2, 7, 15, and 60 days after ensiling and at 4 and 8 days after aerobic exposure. The results showed that dry matter content, crude protein content, the number of lactic acid bacteria, and lactic acid and acetic acid content increased and pH and ammonia nitrogen content decreased after lactic acid bacteria (LAB) inoculation compared with the control group (CK). LP had the lowest pH and highest lactic acid content but did not have greater aerobic stability. LB maintained a lower pH level and acetic acid remained at a higher level after aerobic exposure; aerobic bacteria, coliform bacteria, yeast, and molds all decreased in number, which effectively improved aerobic stability. The effect of the compound addition of LAB was in between the two other treatments, having higher crude protein content, lactic acid and acetic acid content, lower pH, and ammonia nitrogen content. At the phylum level, the dominant phylum changed from Proteobacteria to Firmicutes after ensiling, and at the genus level, *Lactiplantibacillus* and *Lentilactobacillus* were the dominant genera in both LAB added groups, while *Limosilactobacillus* was the dominant genus in the CK treatment. In conclusion, the addition of LAB can improve native grass silage quality by changing bacterial community structure. LP is beneficial to improve the fermentation quality in the ensiling stage, LB is beneficial to inhibit silage deterioration in the aerobic exposure stage, and compound LAB addition is more beneficial to be applied in native grass silage.

## 1. Introduction

The current rapid development of the livestock industry has led to increasing demand for forage, and abundant natural grassland resources in China can provide forage sources to meet demand. Grazing and mowing are the main ways in which native grass is used [1], and mown forage can be made into hay or used for silage. Hay modulation is currently the main storage utilization method in grassland areas. However, the hay preparation process is greatly affected by weather, and hay preparation often overlaps with the rainy season, when rainfall during the drying process can cause significant nutrient loss and degradation of hay quality [2]. There is also an imbalance in the seasonal supply of native grass. Native grasses grow well from June to September, during which time animals are usually able to obtain sufficient nutrition and increase their body weight [3]. In spring and winter, however, insufficient forage supply and reduced quality of stored hay limit animal production, and some animals lose weight or even die due to insufficient forage [4].

This problem can be effectively alleviated by the preparation of native grass into silage. Ensiling grass can effectively preserve the nutritional value of forage while allowing livestock to eat fresh green and juicy forage even in the cold winter months [5]. During native grass silage preparation, epiphytic lactic acid bacteria (LAB) convert water-soluble carbohydrates (WSC) into lactic acid under anaerobic conditions. This lowers the pH, inhibits the activity of harmful microorganisms, and better preserves the nutritional value of the feed [6]. In general, it is difficult to prepare high-quality silage through direct ensiling due to the low water content, WSC content, and LAB population of native grass [7]. LAB strains have been increasingly used to preserve the silage quality and increase feed utilization rates in recent years [8,9,10]. Homofermentative and heterofermentative LAB inoculation has commonly been used to improve the fermentation quality of silage [11,12,13]. The product of homolactic fermentation is lactic acid, which promotes silage fermentation, rapidly lowers the pH of silage, and inhibits the growth of harmful microorganisms [14,15]. However, some studies have shown that inoculation with homofermentative LAB increases the risk of aerobic spoilage [16], because increased lactic acid can be oxidized easily by yeasts and other harmful microorganisms when the silage is exposed to air [17]. The addition of heterofermentative LAB has attracted attention. The products of heterogeneous fermentation produce ethanol, acetic acid, and carbon dioxide in addition to lactic acid, thus consuming more energy, but the acetic acid produced is effective in inhibiting the growth of harmful microorganisms [18,19]. Li et al. [20] evaluated the effect of *Lactobacillus buchneri* on whole crop maize silage and found that *Lactobacillus buchneri* could improve the aerobic stability of whole crop maize silage and reduce the risk of aerobic degradation. Silva et al. [21] added *Lactobacillus buchneri* to high-moisture maize silage and showed that *Lactobacillus buchneri* significantly improved its aerobic stability. Most current studies on native grass silage have focused on the effect of additives on silage quality and microbial community succession during the fermentation stage. Li et al. [22] used *Lactobacillus plantarum* and molasses additives for native grass silage and found that the additives both improved the quality of the silage to varying degrees and changed the succession of microbial communities. However, there have been fewer studies on the effects of different types of LAB on the nutritional quality, fermentation quality, and microbial community changes, and their interactions during the ensiling and aerobic exposure periods of native grass.

Therefore, the objective of this study was to evaluate the effects of the addition of *Lactobacillus plantarum*, *Lactobacillus buchneri,* and their combinations on the nutritional quality, fermentation quality, and microbial community dynamics during ensiling of native grass and the aerobic exposure phase, and to improve the understanding of the fermentation patterns of native grass.

## 2. Materials and Methods

### 2.1. Study Sites and Silage Preparation

The native grass was mown from a typical grassland (44°12′ E, 116°28′ N) at Maodeng pasture in Xilinguole League, Inner Mongolia, China on 1 August 2021. The dominant species in the native grass were *Leymus chinensis* (Trin.) Tzvel. and *Stipa grandis* P. Smirn. The harvested native grass was cut short to 2–3 cm length and mixed well. The following treatment groups were established: no additive control (CK), *Lactobacillus plantarum* (LP), *Lactobacillus buchneri* (LB), and a combination of *Lactobacillus plantarum* and *Lactobacillus buchneri* (PB). The number of LAB added to the group was 1 × 10^6^ colony-forming units (cfu)/g of FW (all provided by Shandong Zhongke Jiayi Biological Engineering Co., Ltd., Weifang, China), and an equal volume of distilled water was added to the CK treatment. The additives were mixed well with the native grass, and the treated native grass was sealed in polyethylene vacuum bags (22 cm × 32 cm). Each packet of silage was filled with 250 g of native grass, and three replicates were set up for each treatment. All samples were stored at room temperature. Fermentation characteristics, chemical composition, microbial counts, and microbial communities were analyzed after 2, 7, 15, and 60 days of silage fermentation and after 4 and 8 days of aerobic exposure.

### 2.2. Chemical Composition

For chemical analysis, the pre-ensiled native grass and silage samples were dried at 65 °C for 48 h to a constant weight to determine dry matter (DM) content [5]. Water-soluble carbohydrate (WSC) content was determined using the anthrone method [23]. Crude protein (CP) content was computed by multiplying TN content by 6.25 [24]. Acid detergent fiber (ADF) and neutral detergent fiber (NDF) were measured according to the method by Van Soest et al. [25] using an ANKOM fiber analyzer (Model: A2000i; Beijing Anke Borui Technology Co., Ltd., Beijing, China). Crude ash content was determined by burning 2 g of dried sample in a muffle furnace (Model: SX2-10-12N; Shanghai Yiheng Technology Co., Ltd., Shanghai, China), at 550 °C for 5 h [24]. Crude fat (EE) was measured using an ANKOM fat analyzer (Model: XT15i; Beijing Anke Borui Technology Co., Ltd., Beijing, China).

### 2.3. Fermentation Composition

The liquid extract was obtained by taking 10 g of silage sample, adding 90 mL of distilled water, tapping it for 2 min on a homogenizer tapper, and filtering it through four layers of coarse cotton cloth and filter paper. The prepared filtrates were used for measuring pH, ammonia nitrogen (NH_3_-N), and organic acids. The pH was determined by a pH meter (Model: LEICI pH S-3C, Shanghai Yitian Scientific Instrument Co., Ltd., Shanghai, China). The content of lactic acid (LA) and acetic acid (AA) was measured by high performance liquid chromatography (Model: Waters e2695, Milford, MA, USA). The concentration of ammonia nitrogen (NH_3_-N) was measured by the phenol-hypochlorous acid colorimetric method of Broderick and Kang [26].

### 2.4. Aerobic Stability

In order to evaluate the aerobic stability, the samples opened after 60 days of ensiling were placed in a 1 L sterile plastic bottle. A multichannel temperature recorder (Model: MDL-1048A; Shanghai Tianhe Automation Instrument Co., Ltd., Shanghai, China) was inserted in the center of the bottle to record temperature changes. Aerobic stability was defined as the time it took for the temperature in the silage masses to rise 2 °C above ambient temperature [27].

### 2.5. Microbial Counting and Sequencing

Ten grams of fresh native grass or silage was taken and 90 mL of sterile distilled water was added and treated in a homogenizer tapper for 2 min to obtain the bacterial liquid. Serial dilutions were made. The number of LAB, aerobic bacteria (AB), coliform bacteria, yeast, and molds of fresh native grass and silage were determined by the plate count method [28], and expressed as cfu/g of FW. Lactic acid bacteria were cultured using MRS medium, molds and yeasts were cultured using potato dextrose agar medium, aerobic bacteria were cultured using nutrient agar medium, and coliforms were cultured using eosin-methylene blue agar medium. The culture media were from the same manufacturer (Guangzhou Huankai Microbial Science and Technology Co., Ltd., Guangzhou, China).

Native grass silage samples after 2 days (CK2d, LP2d, LB2d, and PB2d), 7 days (CK7d, LP7d, LB7d, and PB7d), 15 days (CK15d, LP15d, LB15d, and PB15d), and 60 days (CK60d, LP60d, LB60d, and PB60d) of ensiling, after aerobic exposure for 4 days (CK4d, LP4d, LB4d, and PB4d) and 8 days (CK8d, LP8d, LB8d, and PB8d), and fresh samples (YL) were tested by Beijing Biomarker Technologies Co. Microbial DNA was extracted from native grass samples using the TGuide S96 Magnetic Bead Extraction Kit (Model: DP812, Tiangen Biochemical Technology (Beijing) Co., Ltd., Beijing, China) according to the manufacturer’s protocol. The PCR products were tested for integrity by electrophoresis using 1.8% agarose (manufacturer: Beijing Bomifuxin Technology Co., Ltd., Beijing, China). Primers 27F (5′-AGRGTTTGATYNTGGCTCAG-3′) and 1492R (5′-TASGGHTACCTTGTTASGACTT-3′) were used to amplify the V3-V4 high variant region of the bacterial 16S rRNA gene. PCR reactions were performed using the following procedure: 95 °C for 2 min; 25 cycles of 98 °C for 10 s, 55 °C for 30 s, and 72 °C for 90 s, with a final extension of 72 °C for 2 min. Amplification products were subjected to concentration (Qubit) and band (agarose gel electrophoresis) detection, and samples that met the conditions were mixed. The reactions were performed on a PCR instrument, and the libraries were purified and recovered using AMpure PB magnetic beads to obtain the online libraries, and the libraries were bound using the PacBio Binding Kit (Pacbio, Menlo Park, CA, USA), Primer (Pacbio, Menlo Park, CA, USA) and Polymerase (Pacbio, Menlo Park, CA, USA). The final reaction products were purified by AMpure PB Beads (Pacbio, Menlo Park, CA, USA) and sequenced on a Sequel II (Pacbio, Menlo Park, CA, USA) sequencer.

### 2.6. Statistical Analyses

The fermentation characteristics and microbial quantity of silage under silage to aerobic conditions were analyzed using two-way ANOVA in SAS 9.4 (SAS Institute, Inc., Cary, NC, USA). The statistical model is as follows:$$Y_{ıjh}=\mu +\alpha_{ı}+\beta_{j}+\alpha \beta_{ıj}+\epsilon_{ıjh}$$
where *Y_ıjh_* is an observation, *μ* is the overall mean, *α_ı_* is the effect of additives (*ı* = CK, LP, LB, PB), *β_j_* is the number of ensiling days (*j* = 2, 7, 15, 60, B4, B8), *αβ_ıj_* is the additives × a number of ensiling days interaction, and *ϵ_ıjh_* is the error. Duncan’s multiple comparisons were used for testing, and statistical significance was at the level of *p* < 0.05 level. Bioinformatic analysis was performed using the microbial diversity analysis platform at https://international.biocloud.net accessed on 1 January 2023.

## 3. Results

### 3.1. Chemical Composition and Microbial Community Structure before Ensiling

The chemical parameters and microbial composition of native grass before ensiling are shown in Table 1. The DM content of the raw materials was 51.12%. ADF, NDF, CP, EE, Ash, and WSC were 33.92, 67.95, 8.31, 2.74, 5.32, and 6.38% DM, respectively. Microbial compositions in the native grass for lactic acid bacteria, aerobic bacteria, yeasts, and molds were 2.81, 7.32, 3.84, and 3.62 log_10_ cfu/g FM, respectively. Fresh native grass coliform bacteria were below the detectable level.

### 3.2. Effect of Additives and Days of Anaerobic Fermentation and Aerobic Exposure on Chemical Parameters of Native Grass Silage

The effects of additives and days of anaerobic fermentation and aerobic exposure on chemical parameters of native grass silage are shown in Table 2. The additive treatments, silage days, and aerobic exposure days had significant effects on DM, ADF, NDF, CP, and WSC content, but had no significant effect on EE or ash content, and their interaction only significantly affected ADF, NDF, and WSC. The DM content showed a decreasing trend during the silage period and aerobic exposure phase. After 60 days of ensiling, CK had the lowest DM content, which was significantly lower than LB (*p* < 0.05). After 4 days of aerobic exposure, DM content in LB was significantly higher than that in the other groups (*p* < 0.05). After 8 days of aerobic exposure, there were no significant differences among treatment groups. CP and WSC content showed a tendency to decrease in both the silage and aerobic exposure stages. After 60 days of ensiling, the CP content of LP, LB, and PB was higher than that of CK, among which PB was significantly higher than CK (*p* < 0.05). After 8 days of aerobic exposure, the CP content of LP decreased to lower than that of the CK treatment. The decrease in WSC content in LB was the most significant. After 60 days of ensiling, WSC content in LB was significantly lower than that in CK and LP (*p* < 0.05), and after 8 days of aerobic exposure, WSC content in LB was significantly lower than that in CK, LP, and PB (*p* < 0.05). ADF and NDF showed a decreasing trend during the silage period and an increasing trend during the aerobic exposure period. After 60 days of silage, there was no significant difference among the treatments.

### 3.3. Effect of Additives and Days of Anaerobic Fermentation and Aerobic Exposure on Fermentation Quality of Native Grass Silage

The effects of additives and days of anaerobic fermentation and aerobic exposure on the fermentation quality of native grass silage are shown in Table 3. Additive treatments, silage days, and aerobic exposure days had significant effects on pH, lactic acid, acetic acid, and NH_3_-N content, and their interaction significantly affected pH, lactic acid, and acetic acid. The pH value showed a decreasing trend during silage fermentation and an increasing trend during aerobic exposure. From the seventh day of ensiling, CK continuously had a significantly higher pH than the other groups. After 60 days of ensiling, the pH value of LP was significantly lower than that of CK and LB (*p* < 0.05). pH increased during aerobic exposure, and after 8 days of aerobic exposure, the pH value of LB was lower than that of CK, LP, and PB. The lactic acid content showed an increasing trend during silage fermentation and a decreasing trend during aerobic exposure. After 60 days of ensiling with aerobic exposure for 8 days, the lactic acid content of LP and PB were significantly higher than those of CK and LB (*p* < 0.05), and the lactic acid content of LB was significantly higher than that of CK (*p* < 0.05). Acetic acid content showed an increasing trend during silage fermentation and a decreasing trend during aerobic exposure. After 60 days of ensiling, LB had the highest acetic acid content, and LP, LB, and PB had significantly higher acetic acid content than CK (*p* < 0.05), while LP had significantly lower acetic acid content than LB and PB (*p* < 0.05). After experiencing a reduction in the aerobic exposure stage, the acetic acid content of all treatment groups was significantly reduced, but on the eighth day of aerobic exposure, the acetic acid content of LP, LB, and PB were still significantly higher than that of CK (*p* < 0.05). However, the difference between LP, LB, and PB was not significant, and the acetic acid content of LB was still the highest. NH_3_-N content showed an increasing trend during silage fermentation and aerobic exposure. After 60 days of ensiling, NH_3_-N content was highest in CK and lowest in PB, but the difference between the groups was not significant. NH_3_-N content continued to increase during the aerobic exposure phase, and the NH_3_-N content of CK was significantly higher than the other groups on day 4 of aerobic exposure (*p* < 0.05). On day 8 of aerobic exposure, the NH_3_-N content of LP rose to the highest level, which was significantly higher than that of PB (*p* < 0.05).

### 3.4. Effect of Additives and Days of Anaerobic Fermentation and Aerobic Exposure on Microorganism Counts of Native Grass Silage

The effects of additives and days of anaerobic fermentation and aerobic exposure on microorganism counts of native grass silage are shown in Table 4. The additive treatments, silage days, and aerobic exposure days and their interactions had significant effects on microorganism counts. After 60 days of silage fermentation, the numbers of lactic acid bacteria in LP, LB, and PB were significantly higher than CK (*p* < 0.05). LB was significantly higher than the other groups. After aerobic exposure, except for the increase of lactic acid bacteria in CK, the other groups showed different degrees of decrease. After 8 days of aerobic exposure, the lactic acid bacteria in LP decreased to the lowest level, and the lactic acid bacteria in CK were significantly higher than those in the other groups (*p* < 0.05). The populations of aerobic bacteria and yeasts tended to decrease during silage fermentation and to increase during aerobic exposure. After 60 days of silage fermentation, the aerobic bacterial counts were significantly lower in the CK group than in the other groups. After the elevated aerobic exposure phase, the aerobic bacterial counts were significantly higher in CK than in all the remaining groups on day 8 of aerobic exposure (*p* < 0.05). Coliform bacteria were present only in the pre-silage and late aerobic exposure periods, and were only found in the CK treatment after 15 days of ensiling, while all treatments were free of coliform bacteria after 60 days of silage fermentation. CK produced coliform bacteria on day 4 of aerobic exposure and by day 8 of aerobic exposure, all treatments produced coliform bacteria. CK had the highest number of coliform bacteria, which was significantly higher than LP (*p* < 0.05). Molds were found only in CK, where the number of molds gradually decreased during the pre-silage period and decreased to undetectable levels after 7 days of silage fermentation. After an elevated aerobic exposure phase, the mold in CK was as high as 7.01 log cfu/g FM on day 8 of aerobic exposure.

### 3.5. Effect of Lactic Acid Bacteria Additives on Aerobic Stability during Aerobic Exposure

The effect of lactic acid bacteria additives on the aerobic stability of native grass silage is shown in Figure 1. The aerobic stability of native grass silage under CK, LP, LB, and PB treatments were 106 h, 124 h, 288 h, and 212 h, respectively. The aerobic stability of LB and PB were significantly higher than those of CK and LP (*p* < 0.05), and LB was significantly higher than that of PB (*p* < 0.05). The aerobic stability of each group in descending order was: LB, PB, LP, and LB. Although the aerobic stability of LP was higher than that of CK, the difference was not significant.

### 3.6. Effect of Additives and Days of Anaerobic Fermentation and Aerobic Exposure on Microbial Alpha Diversity of Native Grass Silage

The effects of additives and days of anaerobic fermentation and aerobic exposure on the microbial alpha diversity of native grass silage are shown in Table 5. The additive treatments had significant effects on the OTU, Shannon, and Simpson indexes. Days of silage fermentation and aerobic exposure had significant effects on the OTU, Chao1, Shannon, Simpson, and ACE. Their interaction had significant effects only on the Chao1, Shannon, and Simpson indexes. The average Good’s coverage for all of the samples was greater than 99%, indicating that the sequencing depth was sufficient to capture most bacterial communities in all silages. In terms of the OTU and Chao1 indicators, the abundance of bacterial communities in CK silage initially showed an increasing trend and then a decrease with longer ensiling duration, and there was a decreasing trend with increasing aerobic exposure time. On both day 7 and day 15 of silage fermentation, CK had significantly higher OTU than LP, LB, and PB (*p* < 0.05), and Chao1 was also higher than LP, LB, and PB. After 60 days of silage fermentation, OTU and Chao1 were higher in LP and LB than in CK and PB. After eight days of aerobic exposure, LP and LB had significantly higher OTUs than CK and PB (*p* < 0.05), and Chao1 was also higher than for both CK and PB. Shannon, Simpson, and ACE showed similar trends to those of OTU and Chao1.

### 3.7. Effect of Additives and Days of Anaerobic Fermentation and Aerobic Exposure on Microbial Community Dynamics of Native Grass Silage

This study determined phylum and genus-level changes in the microbial community of native grass silage and differences in bacterial taxa (Figure 2). The phylum-level bacterial community, after the silage of native grass, is shown in Figure 2A. Compared to fresh samples, Firmicutes replaced Proteobacteria as the dominant phyla after silage fermentation. During the silage process, the bacterial phyla in the CK group were more abundant than those in the additive groups, and the abundance of Firmicutes in the additive groups was significantly higher than that in the CK group. In the additive groups, the abundance of Firmicutes in the LP group was higher in the pre-middle stage of ensilage, while that in the LB group was higher in the late stage of ensilage. During aerobic exposure, the abundance of Firmicutes in each treatment group initially increased slightly, and decreased significantly in CK and LP groups at the later stage.

The genus-level changes of the bacterial community are shown in Figure 2B. In the fresh samples, *Pantoea* was the main genus, followed by *Pseudomonas*. At 2 days of silage fermentation, *Lactiplantibacillus* were the most abundant bacteria in CK, LP, and PB groups, and *Lactiplantibacillus* and *Lentilactobacillus* were the most abundant bacteria in the LB group. At 7 days of ensiling, *Enterobacter*, *Pantoea*, and *Levilactobacillus* were the most abundant bacteria in the CK group, while *Lactiplantibacillus* was the most abundant bacteria in the LP group, followed by *Lentilactobacillus*. *Lentilactobacillus* was higher in the LB group, and *Lentilactobacillus* was higher in the PB group, followed by *Lactiplantibacillus*. At 15 days of ensiling, *Enterobacter* and *Escherichia-Shigella* were the most abundant bacteria in the CK group, *Lactiplantibacillus* was the most abundant in the LP group, and *Lentilactobacillus* was the most abundant in the LB and PB groups. At 60 days of ensiling, *Limosilactobacillus* was the most abundant bacteria in the CK group, *Lactiplantibacillus* was the most abundant bacteria in the LP group, and *Lentilactobacillus* was the most abundant bacteria in the LB and PB groups. During aerobic exposure, *Limosilactobacillus* and *Levilactobacillus* were the main bacteria in the CK group, while *Lentilactobacillus* were the main bacteria in the additive groups after 4 days of aerobic exposure. Bacteria in LB and PB were mainly *Lactiplantibacillus*. After 8 days of aerobic exposure, the main bacterial flora of the CK group changed to *Levilactobacillus* and *Enterobacter*, while the bacterial species of the LP group were significantly increased and the abundance of *Lentilactobacillus* of the LB and PB groups remained at a high level.

The results of the LEfSe analysis are shown in Figure 2C, which indicates that the bacterial communities differed among treatments. In the YL group, the Class Gammaproteobacteria, Phylum Proteobacteria, Order Enterobacterales, Family Erwiniaceae, and Genus *Pantoea agglomerans* had the greatest effect on the difference between groups. In the PB group, the Class Bacilli, Phylum Firmicutes, Family Lactobacillaceae, and Order Lactobacillales had the greatest effect on intergroup differences. In the LP group, the *Lactiplantibacillus plantarum* and Genus *Lactiplantibacillus* had the greatest effect on the difference between groups. In the LB group, the *Lentilactobacillus buchneri* and Genus *Lentilactobacillus* had the greatest effect on the difference between groups. In the CK group, the Family Enterobacteriaceae, Genus *Limosilactobacillus,* and *Limosilactobacillus fermentum* had the greatest effect on the difference between groups.

### 3.8. Correlation of Microbial Genera Level with Silage Quality in Native Grass Silage

The correlations between bacterial genus levels and silage quality during silage fermentation and aerobic exposure are shown in Figure 3. *Lentilactobacillus* was extremely significantly positively correlated with lactic acid (*p* < 0.01), and significantly positively correlated with ADF (*p* < 0.05), but was extremely significantly negatively correlated with pH and WSC (*p* < 0.01). *Massilia* was extremely significantly positively correlated with acetic acid (*p* < 0.01), but was extremely significantly negatively correlated with WSC (*p* < 0.01). *Lactiplantibacillus* was significantly positively correlated with DM, WSC, and CP (*p* < 0.05), but was extremely significantly negatively correlated with NH_3_-N and ADF (*p* < 0.01). *Blautia* was extremely significantly positively correlated with EE (*p* < 0.01). *Pantoea* was extremely significantly negatively correlated with ADF (*p* < 0.01), and significantly negatively correlated with NH_3_-N and lactic acid (*p* < 0.05). *Enterobacter* was significantly positively correlated with pH (*p* < 0.05). *Levilactobacillus* was significantly positively correlated with pH (*p* < 0.05), but was significantly negatively correlated with DM and acetic acid (*p* < 0.05). *Acinetobacter, Enhydrobacter, Staphylococcus, Cutibacterium,* and *Phascolarctobacterium* were significantly positively correlated with NH_3_-N (*p* < 0.05).

## 4. Discussion

The feedstock characteristics of silage directly affect the fermentation quality of silage [29], especially the number of LAB attached to the silage feedstock and the WSC content. Successful silage is more likely to be produced when the feedstock contains more than 5.0 log cfu/g FW of LAB and more than 5% DM of WSC [30]. In this study, the WSC content was 6.38% DM, but the LAB content in the feedstock was only 2.81 log cfu/g FW, which was much lower than 5.0 log cfu/g FW. Therefore, it is difficult to produce high-quality silage by direct ensiling of native grass, and it is necessary to add exogenous LAB additives to promote lactic acid fermentation and produce high-quality silage.

As shown in Table 2, DM decreased in small amounts during silage fermentation and decreased rapidly during the aerobic exposure phase. The DM loss in the LAB additive group was less than that in the control group at all stages, and the DM loss in LB was less than that in all other groups, which is in agreement with the results of Avila et al. [31]. This effect was caused by the inoculated strains becoming dominant during silage fermentation, which suppressed parthenogenic aerobic microorganisms growing at the beginning of silage fermentation, including those that can survive throughout the fermentation process as well as harmful microorganisms produced during the aerobic exposure phase [32], such as yeasts, molds, etc. The addition of LAB to silage allows more efficient fermentation of WSC to produce lactic acid, resulting in a significant decrease in pH and reduced nutrient losses [33], so that the WSC content was lower in all treatments after fermentation with the addition of LAB than in the CK treatment. However, the WSC of CK was lower than that of LP after 8 days of aerobic exposure due to the increase in temperature in CK and the presence of eye-visible mold, which accelerated the degradation of WSC, while the conversion of starch to hemicellulose led to an increase in WSC for the LP group, which is consistent with the findings of Tabacco et al. [34]. During ensiling of native grass, CP content was significantly reduced, which was mainly due to the degradation of proteins by some microorganisms involved in silage [7]. After 60 days of ensiling, the CP content of all treatments with LAB addition was higher than conventional silage, and the combination of *Lactobacillus plantarum* and *Lactobacillus buchneri* had significantly higher CP than conventional silage, indicating that the addition of LAB could inhibit the degradation of protein by undesirable microorganisms, while the effect of *Lactobacillus buchneri* was better than that of *Lactobacillus plantarum* and their combined application was even more effective, which is consistent with the results of Tabacco et al. [34] and Si et al. [32]. After experiencing aerobic exposure, CP content showed different degrees of decrease in all treatments, and after 8 days of aerobic exposure, the CP of PB remained the highest, while the CP of LP decreased to lower than that of CK, because LP produced a large amount of lactic acid during silage fermentation, which provided a large amount of fermentation substrate for harmful microorganisms after aerobic exposure, which enabled them to multiply and accelerated the degradation of CP. Therefore, LP has a detrimental effect on change in nutritional quality during the aerobic exposure phase of silage.

The pH of silage is a key indicator to evaluate the quality of fermentation, and the pH of well-fermented silage should be 4.2 or lower [17]. In the present study, the pH of all groups decreased with longer ensiling duration. The lactobacilli addition group had lower pH than CK at all stages, and after 60 days of ensiling, the pH of the LP group was the lowest, close to the 4.2, due to the large amount of lactic acid produced by *Lactobacillus plantarum* in silage, resulting in a rapid decrease in pH, as was also shown by Zi et al. [10] and Ren et al. [35]. *Lactobacillus buchneri* was slightly less effective in reducing the decrease in pH than Lactobacillus plantarum, with a pH of 4.72 after 60 days of silage. However, after aerobic exposure, the pH of the Lactobacillus plantarum addition group increased rapidly and was already higher than *Lactobacillus buchneri* at day 8 of aerobic exposure. The increase in pH and NH_3_-N content was slowed down by *Lactobacillus buchneri* compared to *Lactobacillus plantarum*, confirming the positive effect of LB on improving the aerobic stability of silage, which is similar to results reported by Huang et al. [36]. pH increase was due to the utilization of organic acids as substrate by yeasts, molds, or aerobic bacteria in general under aerobic conditions, or organic acid concentration due to reduced volatilization [37], and after 8 days of aerobic exposure, the pH of CK was as high as 6.04 with high numbers of general aerobic bacteria, yeasts, and molds, and produced mold visible to the naked eye with an unpleasant odor. This also indicates the need for the addition of LAB to native grass silage. The addition of Lactobacillus plantarum produced a significant amount of lactic acid, significantly higher than the CK and *Lactobacillus buchneri* addition groups, with the same effect after aerobic exposure, indicating the high-quality fermentation effect of Lactobacillus plantarum. It has been shown that acetic acid is derived from the breakdown and fermentation of sugars by, for example, heterofermentative LAB [38]. In this study, after 60 days of ensiling, the acetic acid content of the *Lactobacillus buchneri* addition group was significantly higher than that of the CK and *Lactobacillus plantarum* addition groups, and the WSC content was significantly lower than that of the CK and *Lactobacillus plantarum* addition groups, which also verified this notion. NH_3_-N content tended to increase during aerobic exposure to silage, and after 60 days of ensiling, the CK group had higher NH_3_-N content than the Lactobacillus addition group. The combined application of *Lactobacillus plantarum* and *Lactobacillus buchneri* had the lowest NH_3_-N content and the best inhibition of protein degradation. The NH_3_-N content of the *Lactobacillus plantarum* addition group was higher than that of CK after aerobic exposure, further validating the unfavorable effect of *Lactobacillus plantarum* in the aerobic exposure phase, as it can be used as a growth substrate for spoilage-bearing yeasts and molds [39].

Before ensiling, LAB, aerobic bacteria, coliforms, molds, and yeasts are frequently found in native grass [22]. During the silage process, harmful bacteria decreased due to the lowering of pH and the high production of organic acids that inhibited their activity, resulting in lower numbers of harmful bacteria. The presence of coliforms is detrimental to silage production because they compete with LAB for available sugars and degrade proteins [40]. In the present study, coliforms were present only at and before 7 days of ensiling due to a rapid decrease in pH in the LAB addition group, which inhibited the growth of coliforms, while coliforms were present in CK until 15 days of ensiling. After the aerobic exposure stage, when oxygen entered the silage, the increase in pH started to enhance coliform activity. After 4 days of aerobic exposure, coliforms appeared again in CK, and after 8 days of aerobic exposure, coliforms appeared in all groups, but CK had the highest value and the most serious deterioration. A large number of LAB is required in silage to ensure rapid and effective fermentation of silage, which is a key factor in ensuring silage success [41]. LAB counts were all significantly higher in the LAB addition group than in CK, ensuring a more efficient fermentation. After aerobic exposure, the number of harmful bacteria soared, with aerobic bacteria, yeast, and mold counts in CK all higher than 5.0 log cfu/g FW and yeast counts as high as 7.01 log cfu/g FW. The number of harmful bacteria in the LAB addition group was all lower than in CK, thus the quality effect of the LAB additive was considered valuable [41].

High concentrations of acetic acid inhibit the growth of undesirable microorganisms during anaerobic fermentation and improve the aerobic stability of silage, as verified by the aerobic stability shown in Figure 1. The aerobic stability of LB and PB was significantly greater than that of CK and LP, with LB being the highest. Better fermentation characteristics and aerobic stability during the aerobic exposure phase are common when *Lactobacillus buchneri* is used as a silage inoculant, which is consistent with observations reported by Agarussi et al. [42]. While the acetic acid content of LP was higher than that of CK, aerobic stability was not significantly different from CK, which may be due to the fact that a large amount of lactic acid provided sufficient fermentation substrate for undesirable microorganisms during the aerobic exposure phase, resulting in reduced aerobic stability of LP, as verified by the results of Da et al. [43] and Mugabe et al. [44].

In conclusion, these findings suggest that LAB additives can improve the quality of native grass silage fermentation and reduce the number of harmful microorganisms. The LB effect was optimal in the aerobic exposure phase, and LP even played a counterproductive role.

In this study, 16SrRNA sequencing was used to reveal the bacterial diversity and composition of native grass silage in anaerobic fermentation versus aerobic exposure with or without different LAB inoculation. All samples had coverage values greater than 0.99, indicating that the sequencing range was quite broad and that microbial high-throughput data were sufficient to define the characteristics of the bacterial microbial community [45]. The OTU and Chao1 values of the LAB addition group were lower than those of CK on and before 15 days of ensiling, indicating that the addition of LAB inhibited the growth of miscellaneous bacteria in the pre-silage period and exogenous LAB gradually became dominant, resulting in lower OTU and Chao1 values of the LAB addition group than CK, which is consistent with the findings of Yan et al. [46] and Mu et al. [47]. After 60 days of ensiling, the OTU and Chao1 values of the lactobacilli-addition groups was higher than those of CK. Similar effects were found by Xiong et al. [48] after the addition of Lactobacillus propionic and Lactobacillus paracasei to oat silage, which may be due to the fact that the pasture itself began to proliferate with attached lactobacilli after the inhibition and disappearance of miscellaneous bacteria in the first stage of silage fermentation, and the addition of exogenous lactobacilli stimulated the competition of attached lactobacilli, resulting in higher OTU and Chao1 values of the lactobacilli added group than CK in the later stage of silage. After aerobic exposure, the OTU and Chao1 values in the LP group were higher than the other groups, indicating that LP could not effectively inhibit the growth of various bacteria in the aerobic exposure stage, while the lactic acid produced during aerobic exposure could provide substrate for the growth and reproduction of harmful bacteria, as was also shown in the study by Liu et al. [49]

The abundance analysis of microbial communities showed that Firmicutes replaced Proteobacteria as the dominant phylum after silage fermentation compared with fresh samples. This is because Proteobacteria is the largest phylum of bacteria and is common in various raw silage materials [50], and because Firmicutes contain various LAB and are the main participants in silage fermentation [51], which is consistent with the results of Wang et al. [52]. During the ensiling process, the CK group had a lower abundance of thick-walled bacteria phyla and more phylum species, while the additive groups had a higher abundance of thick-walled bacteria phyla. This is due to the fact that the content of LAB attached to the raw material in the CK group was low, and the inhibition of harmful microorganisms during ensiling was poor [53], while the groups with added LAB were able to enter the lactic acid fermentation stage faster and effectively inhibit the growth and reproduction of miscellaneous bacteria, which is consistent with the results of Kung et al. [54]. Among the additive groups, the abundance of the thick-walled phyla was higher in the LP group in the middle pre-fermentation period and in the LB group in the later period. This may be related to the tolerance of *Lactobacillus plantarum* to the acidic environment, as it can grow normally in the first and middle silage stages, and gradually starts to die when pH is below a certain value, while *Lactobacillus buchneri* has better tolerance and can grow normally in the late fermentation stage [55,56]. During aerobic exposure, the abundance of thick-walled bacterial phyla increased slightly in the initial stage in each treatment group and decreased significantly in the later stages in the CK and LP groups. This was due to the poor inhibition of aerobic microorganisms in the CK and LP groups, leading to a rapid increase in other phyla, which is consistent with the results of Liu et al. [49].

The analysis of microbial abundance at the genus level showed that *Pantoea* was the main genus in fresh samples, followed by *Pseudomonas*, which are common harmful microorganisms on raw silage materials. *Pantoea* are facultative anaerobic bacteria which always exist in the process of silage fermentation, while *Pseudomonas* are aerobic bacteria which will rob nutrients in the early stage of ensiling [57,58]. During silage fermentation, *Lactiplantibacillus* was the dominant bacterial genus in the LP and PB groups, and *Lentilactobacillus* was the dominant bacterial genus in the LB group, due to the addition of *Lactobacillus plantarum* in the LP and PB groups, which was able to promote the lactic acid fermentation process and inhibit the growth of other microorganisms in the first and middle stages, resulting in a higher abundance of the genus *Lactiplantibacillus*, which is consistent with the results of Xian et al. [59]. *Lactobacillus buchneri* was added in the LB and PB groups. It is more acid-tolerant and can grow normally in the late silage period, so the abundance of *Lentilactobacillus* was higher, which is consistent with the results of Tao et al. [60]. The genus composition of the CK group in the pre-silage period is more diverse, which is due to the slow process of lactic acid fermentation, resulting in the growth of more miscellaneous bacteria. Among them, *Enterobacter* and *Escherichia-Shigella* are harmful bacteria that affect silage fermentation, as they are pathogenic and can cause animal diseases that endanger livestock health [61]. The abundance of miscellaneous bacteria decreased significantly and the abundance of *Limosilactobacillus* increased in the later stages of silage fermentation due to the lower pH. The abundance of Lactobacillus remained high in all groups after four days of aerobic exposure, and after eight days of aerobic exposure the predominant flora in the CK group changed to *Levilactobacillus* and *Enterobacter,* while the bacterial species in the LP group increased significantly and the abundance of *Lentilactobacillus* in the LB and PB groups remained at high levels. This is probably because some of the Lactobacillus remained in an anaerobic environment due to the gradual downward infiltration of air in the pre-aerobic exposure period, while some of the Lactobacillus were partly anaerobic, which is consistent with the findings of Keshri et al. [62]. On day 8 of aerobic exposure, the degree of aerobic deterioration was more serious in the CK and LP groups, as the abundance of miscellaneous bacteria such as *Enterobacter* increased again, which is consistent with the results of Liu et al. [63].

LEfSe analysis was used to further analyze the differences in bacterial communities between treatment groups. Among the YL groups, Proteobacteria, *Enterobacter*, and *Pantoea* were the common microorganisms in silage feedstock, and their growth and reproduction were inhibited and abundance decreased significantly after silage fermentation, so they were the differential microorganisms with respect to silage, which is in agreement with the results of Li et al. [64] and Sa et al. [65]. The differential bacteria in the CK group were Enterobacteriaceae and *Limosilactobacillus*. This was due to the slower lactic acid fermentation in the CK group, where Enterobacteriaceae competed with LAB and were in higher abundance during ensiling than in the additive groups [66]. Additionally, *Limosilactobacillus*, which may be bacteria attached to the native grass itself, became the main bacteria guiding the fermentation in the CK group [67]. The differential bacteria in the LP, LB, and PB groups were *Lactiplantibacillus*, *Lentilactobacillus*, and Lactobacillaceae, which corresponded to the LAB added in each group, confirming that the use of LAB additives can change the main flora of silage fermentation, which is consistent with the studies of Chen et al. [68] and Drouin et al. [69].

Correlation analysis between bacterial genus levels and silage quality showed that *Lentilactobacillus* was highly significantly and positively correlated with lactic acid, but highly and negatively correlated with pH and WSC. This may be due to the fact that the type of fermentation of *Lentilactobacillus* is heterotactic and the lactic acid and acetic acid produced by using WSC as a substrate can effectively reduce pH while improving the aerobic stability of silage, which plays an important role during silage and aerobic exposure [70,71]. Massilia is an undesirable genus of bacteria in silage, which affects silage quality and causes loss of WSC, which is in agreement with the results of Ren et al. [35]. *Lactiplantibacillus* was significantly positively correlated with DM, WSC, and CP, but highly significantly negatively correlated with NH_3_-N and ADF, due to the fact that *Lactiplantibacillus* breaks down fiber into soluble carbohydrates and then produces large amounts of lactic acid through isotype fermentation to inhibit the breakdown of dry matter and crude protein by unfavorable microorganisms and reduce the production of ammoniacal nitrogen [72]. *Pantoea* showed a significant negative correlation with NH_3_-N and lactic acid, and *Enterobacter* showed a significant positive correlation with pH, due to the fact that *Pantoea* and *Enterobacter* are harmful microorganisms in silage that break down proteins to produce ammonia nitrogen, but the low pH environment formed by large amounts of lactic acid inhibits their growth and reproduction, which is in agreement with the results of Du et al. [50] and Hu et al. [73]. *Levilactobacillus* showed a significant positive correlation with pH, but a significant negative correlation with DM and acetic acid. This could be due to the poor acid tolerance or competitiveness of *Levilactobacillus*, which cannot grow properly with decreasing pH in the middle and late stages of silage fermentation [74]. *Acinetobacter*, *Enhydrobacter*, *Staphylococcus*, *Cutibacterium*, and *Phascolarctobacterium* were significantly and positively associated with NH_3_-N, which are commonly found in spoiled silage and milk. Their growth metabolism breaks down proteins to produce ammonia nitrogen, affecting the quality and palatability of silage and producing toxins that can cause animal diseases [75,76].

## 5. Conclusions

This study evaluated the effects of *Lactobacillus plantarum* and *Lactobacillus buchneri* and their mixed additions on fermentation parameters, chemical composition, and bacterial communities of native grass silage and aerobic exposure stages. The results of the study showed that all LAB additives could improve the silage quality of native grass silage to varying degrees and regulate its microbial composition. *Lactobacillus plantarum* could produce more lactic acid and lower pH to improve fermentation quality. *Lactobacillus buchneri* can effectively inhibit the growth of harmful microorganisms during aerobic exposure and better improve aerobic stability. Additionally, the mixed addition of *Lactobacillus plantarum* and *Lactobacillus buchneri* combined the advantages of both alone and showed excellent results in both silage and aerobic exposure stages. In conclusion, the compound addition of Lactobacillus is more favorable for application in native grass silage. This finding can lay the theoretical foundation for our future research on compound LAB suitable for producing native grass silage.

## Figures and Tables

**Figure 1 microorganisms-11-00513-f001:**
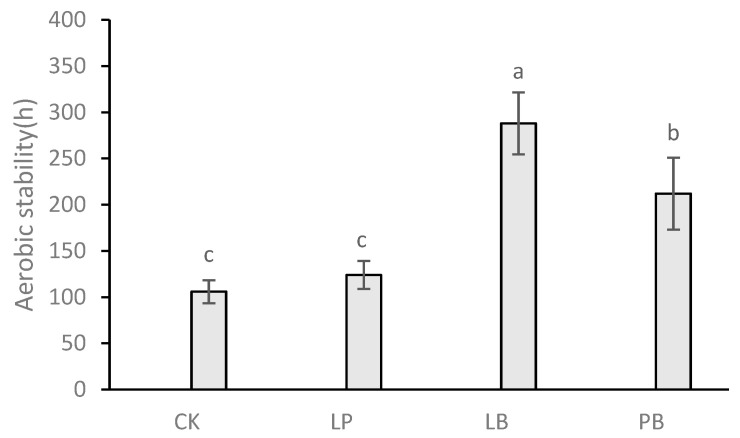
Time required to exceed room temperature by 2 °C during aerobic exposure of native grass silage. CK, control group; LP, *Lactobacillus plantarum*; LB, *Lactobacillus buchneri*; PB, *Lactobacillus plantarum* + *Lactobacillus buchneri*. Different lowercase letters indicate significant differences among different treatments (*p* < 0.05).

**Figure 2 microorganisms-11-00513-f002:**
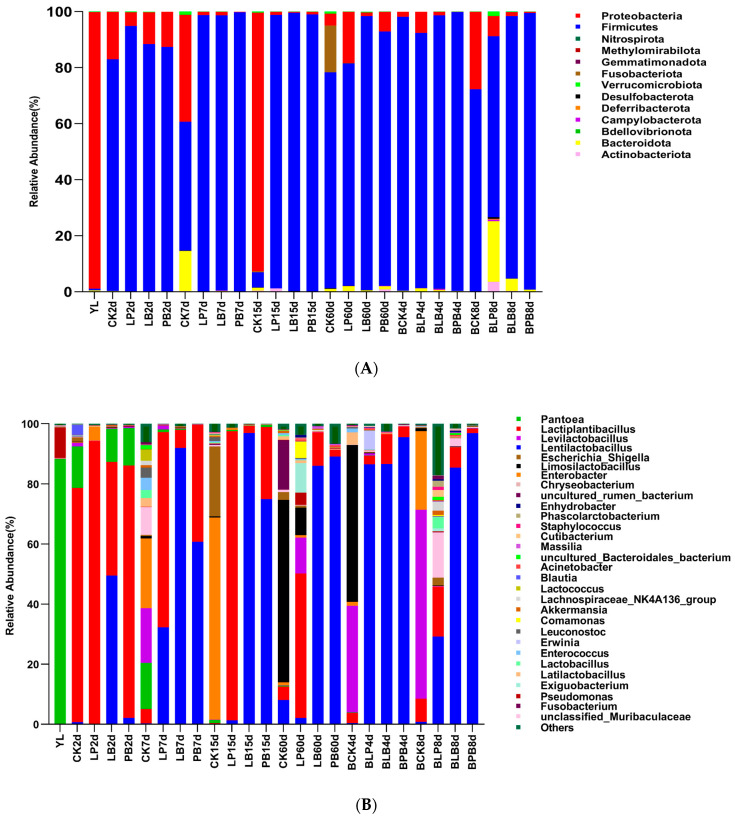
Differences in bacterial taxa and bacterial abundance at phylum and genus level of native grass silage. CK, control group; LP, *Lactobacillus plantarum*; LB, *Lactobacillus buchneri*; PB, *Lactobacillus plantarum* + *Lactobacillus buchneri*. The bacterial communities are shown at the phylum level (**A**) and the genus level (**B**). Differences in bacterial taxa were revealed using LEfSe analysis (**C**).

**Figure 3 microorganisms-11-00513-f003:**
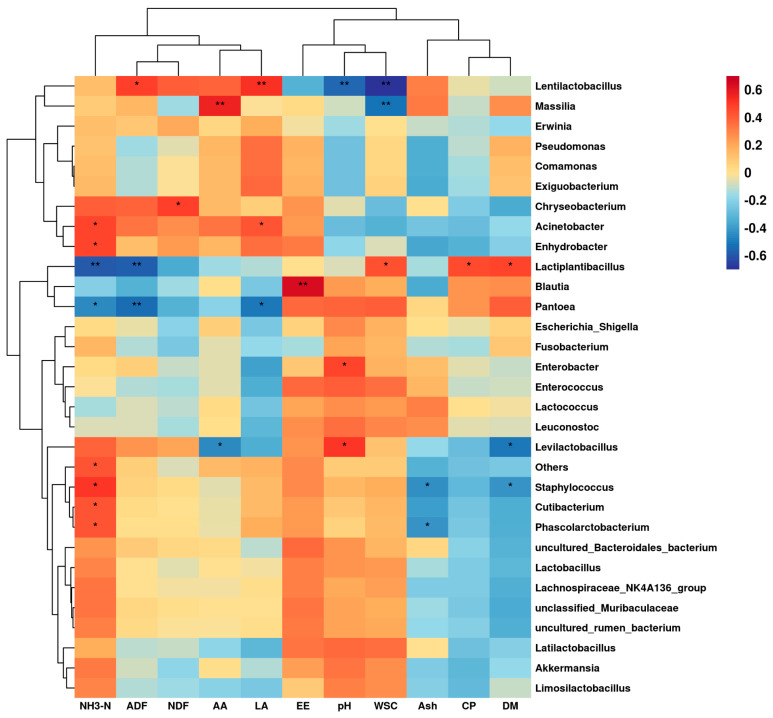
Correlation of bacterial at the genera-level with silage quality during anaerobic fermentation and aerobic exposure of silage. Red represents a positive correlation, while blue represents a negative correlation. *: Significant correlation at *p* < 0.05 level, **: Significant correlation at *p* < 0.01 level.

**Table 1 microorganisms-11-00513-t001:** Chemical and microbial compositions of fresh native grass.

	Items	Sample	SEM
Chemical composition	Dry matter (% FM)	51.12	0.12
	Acid detergent fiber (% DM)	33.92	0.33
	Neutral detergent fiber (% DM)	67.95	0.25
	Crude protein (% DM)	8.31	0.16
	Crude fat (% DM)	2.74	0.07
	Coarse ash (% DM)	5.32	0.09
	Water soluble carbohydrate (% DM)	6.38	0.33
Microbial counts	Lactic acid bacteria (log_10_ cfu/g FM)	2.81	0.23
	Aerobic bacteria (log_10_ cfu/g FM)	7.32	0.12
	Coliform bacteria (log_10_ cfu/g FM)	ND	ND
	Yeasts (log_10_ cfu/g FM)	3.84	0.20
	Molds (log_10_ cfu/g FM)	3.62	0.26

FM, fresh material; DM, dry matter; ADF, acid detergent fiber; NDF, neutral detergent fiber; CP, crude protein; EE, crude fat; Ash, coarse ash; WSC, water soluble carbohydrates; ND, not detected; SEM: standard error of mean.

**Table 2 microorganisms-11-00513-t002:** Effects of additives and days of anaerobic fermentation and aerobic exposure on the chemical composition of native grass silages.

Items	Treatment	Days						Significance			
		2	7	15	60	B4	B8	SEM	T	D	T × D
DM (% FM)	CK	47.68Aa	44.66Ab	45.65Ab	46.01Bb	42.02Bc	40.34Ad	0.28	**	**	NS
	LP	46.95Aa	44.26Ab	45.71Aab	46.54ABa	42.25Bc	41.36Ac				
	LB	47.42Aa	45.13Abc	46.91Aab	47.11Aab	44.63Ac	42.53Ad				
	PB	47.88Aa	44.62Abc	46.33Aab	46.93ABa	43.40ABcd	41.73Ad				
ADF (% DM)	CK	32.87Bc	34.65ABb	35.05ABb	34.28Abc	34.93Cb	37.83Aa	0.20	**	**	**
	LP	34.33Ab	34.52ABb	35.16aABb	34.25Ab	35.89BCa	35.39Bab				
	LB	33.43ABc	34.89Abc	35.92Ab	35.46Ab	37.93Aa	37.92Aa				
	PB	32.27Bd	32.79Bd	34.51Bc	35.74Ab	36.94ABa	37.33Aa				
NDF (% DM)	CK	63.56ABc	64.02Abc	63.46Ac	62.89Ac	65.17Bb	67.17Aa	0.24	**	**	**
	LP	64.80Aab	63.64Ab	64.62Aab	64.78Aab	66.67ABa	65.05Bab				
	LB	64.06ABbc	64.32Abc	66.00Ab	63.32Ac	68.62Aa	68.22Aa				
	PB	62.95Bc	62.23Bc	64.65Abc	62.77Ac	66.60ABab	68.51Aa				
CP (% DM)	CK	8.87ABa	8.28Ba	8.25Aa	7.94Ba	7.67Ba	7.90Aa	0.07	**	**	NS
	LP	8.38ABab	8.64ABa	8.49Aa	7.97Bab	7.98Bab	7.73Ab				
	LB	7.90Bb	9.15ABa	7.64Ab	8.09ABb	7.88Bb	7.93Ab				
	PB	9.02Aab	9.54Aa	8.05Ac	8.34Abc	8.46Abc	8.17Abc				
EE (% DM)	CK	3.99Aa	3.25Aab	2.87Aab	2.62Ab	3.34Aab	2.95Aab	0.07	NS	NS	NS
	LP	2.81Aa	2.73ABa	2.22Aa	3.12Aa	2.79Aa	3.26Aa				
	LB	2.38Ab	2.35Bb	2.63Aab	2.72Aab	2.55Aab	3.23Aa				
	PB	2.89Aa	2.54ABa	2.71Aa	3.01Aa	2.97Aa	2.71Aa				
Ash (% DM)	CK	4.45Aa	5.19Aa	4.95Aa	4.68Ba	4.70Ba	4.65Ba	0.04	NS	NS	NS
	LP	4.83Aa	4.81Aa	4.75Aab	4.48Bbc	4.72Bab	4.38Cc				
	LB	4.58Ab	4.92Aab	5.04Aa	5.12Aa	5.28Aa	5.00Aa				
	PB	5.15Aa	5.13Aa	4.81Ab	4.60Bb	4.89Bab	4.85Aab				
WSC (% DM)	CK	2.98Aa	3.21Aa	2.79Aa	2.90Aa	3.20Aa	1.95Bb	0.11	**	**	**
	LP	2.96Aab	3.35Aa	2.60ABab	2.44Ab	2.27Bb	2.86Aab				
	LB	2.96Aa	2.92Aa	1.51Cb	0.57Bc	0.77Cc	0.89Dc				
	PB	2.79Aa	3.07Aa	2.06BCb	1.28Bc	1.01Cc	1.30Cc				

CK, control group; LP, *Lactobacillus plantarum*; LB, *Lactobacillus buchneri*; PB, *Lactobacillus plantarum* + *Lactobacillus buchneri*; FM, fresh material; DM, dry matter; ADF, acid detergent fiber; NDF, neutral detergent fiber; CP, crude protein; EE, crude fat; Ash, crude ash; WSC, water soluble carbohydrates; SEM, standard error of the mean; T, treatments; D, ensiling and aerobic exposure days; T × D, interaction between T and D; NS, not significant; * Significant at 0.05. ** Significant at 0.01. Different capital letters indicate significant differences among different treatments on the same ensiling and aerobic exposure days (*p* < 0.05); different lowercase letters indicate significant differences among different ensiling and aerobic exposure days under the same treatment (*p* < 0.05); no or same letter indicate not significant (*p* > 0.05).

**Table 3 microorganisms-11-00513-t003:** Effects of additives and days of anaerobic fermentation and aerobic exposure on fermentation quality of native grass silages.

Items	Treatment	Days						Significance			
		2	7	15	60	B4	B8	SEM	T	D	T × D
pH value	CK	5.57ABb	5.58Ab	5.55Ab	5.47Ab	5.62Aab	6.04Aa	0.07	**	**	**
	LP	5.67Aa	4.31Ccd	4.17Cd	4.24Cd	4.53Bc	5.17Bb				
	LB	5.10Ba	4.68Bbc	4.59Bc	4.72Bbc	4.74Bbc	4.92Bab				
	PB	5.32ABa	4.30Ccd	4.20Cd	4.56BCbcd	4.64Bbc	4.98Bab				
Lactic acid (g/kg DM)	CK	0.91ABb	1.18Cb	1.33Bb	3.15Ca	1.15Cb	0.74Cb	0.71	**	**	**
	LP	3.38Ad	7.00Bcd	9.98ABbc	16.73Aa	12.53Aab	11.31Abc				
	LB	0.17Bb	4.00BCab	8.39Ba	8.16Ba	6.33Ba	5.97Ba				
	PB	1.92ABb	12.23Aa	18.20Aa	15.27Aa	14.10Aa	13.63Aa				
Acetic acid (g/kg DM)	CK	7.20Aa	7.43Aa	8.24ABa	6.28Ca	3.37Cb	1.33Bb	0.38	**	**	**
	LP	6.43Abc	4.81Bc	6.60BCbc	9.03Ba	7.53Bab	6.19Abc				
	LB	3.81Ac	1.94Cc	4.40Cc	12.77Aa	11.03Aab	8.85Ab				
	PB	3.89Ac	8.30Aab	10.77Aab	11.54Aa	9.20ABab	7.61Ab				
NH3-N (g/kg DM)	CK	0.24Ad	0.33Acd	0.47Ac	0.71Ab	0.84Aab	0.93ABa	0.03	**	**	NS
	LP	0.23Ac	0.17Ac	0.33Ac	0.65Ab	0.67Bb	1.04Aa				
	LB	0.29Ade	0.17Ae	0.43Acd	0.61Abc	0.68Bb	0.89ABa				
	PB	0.14Ad	0.16Ad	0.36Ac	0.59Ab	0.65Bb	0.78Ba				

CK, control group; LP, *Lactobacillus plantarum*; LB, *Lactobacillus buchneri*; PB, *Lactobacillus plantarum* + *Lactobacillus buchneri*; DM, dry matter; NH_3_-N, ammonia nitrogen; SEM, standard error of mean; T, treatments; D, ensiling and aerobic exposure days; T × D, interaction between T and D; NS, not significant; * Significant at 0.05. ** Significant at 0.01. Different capital letters indicate significant differences among different treatments on the same ensiling and aerobic exposure days (*p* < 0.05); different lowercase letters indicate significant differences among different ensiling and aerobic exposure days under the same treatment (*p* < 0.05); no or same letter indicate not significant (*p* > 0.05).

**Table 4 microorganisms-11-00513-t004:** Effects of additives and days of anaerobic fermentation and aerobic exposure on microorganism counts of native grass silages.

Items	Treatment	Days						Significance			
		2	7	15	60	B4	B8	SEM	T	D	T × D
Lactic acid bacteria (log_10_ cfu/g FM)	CK	6.43Aa	5.3Bb	4.44Bc	2.73Cd	5.11Abc	6.87Aa	0.17	**	**	**
	LP	6.73Aa	7.09Aa	7.24Aa	5.18Bb	2.32Bd	3.56Cc				
	LB	6.71Aa	6.93Aa	7.01Aa	5.71Ab	4.98Ac	4.47BCd				
	PB	6.73Aa	7.08Aa	7.07Aa	4.86Bb	4.41Ab	5.11Bb				
Aerobic bacteria (log_10_ cfu/g FM)	CK	5.77Aab	6.88Aa	5.82Cab	2.58Bc	4.99Ab	6.83Aa	0.18	*	**	**
	LP	6.74Aa	7.08Aa	6.54ABa	4.60Ab	2.62Cd	3.50Bc				
	LB	5.93Ab	7.09Aa	6.24Bab	4.37Ac	4.09Bc	4.05Bc				
	PB	6.62Aa	7.15Aa	6.9Aa	4.42Ab	4.45ABb	4.30Bb				
Coliform bacteria (log_10_ cfu/g FM)	CK	1.31Ab	3.59Aa	3.39a	ND	1.90ab	3.90Aa	0.19	**	**	*
	LP	1.10Aa	ND	ND	ND	ND	1.39Ba				
	LB	1.63Aab	2.98ABa	ND	ND	ND	2.06ABab				
	PB	1.19Ab	1.20ABb	ND	ND	ND	3.31ABa				
Yeast (log_10_ cfu/g FM)	CK	6.62Aa	6.92Ba	6.84Aa	3.18Bc	5.43Ab	6.95Aa	0.16	*	**	**
	LP	6.73Aa	6.91Ba	6.80Aa	4.47Ab	3.10Bc	5.11Bb				
	LB	5.76Aab	6.92Ba	6.89Aa	4.38Ac	4.84Abc	5.82ABab				
	PB	6.91Aa	7.39Aa	6.95Aa	4.42Ab	4.74Ab	5.14Bb				
Molds (log_10_ cfu/g FM)	CK	1.74bc	1.12bc	ND	ND	2.90b	7.01a	0.19	**	**	**
	LP	ND	ND	ND	ND	ND	ND				
	LB	ND	ND	ND	ND	ND	ND				
	PB	ND	ND	ND	ND	ND	ND				

CK, control group; LP, *Lactobacillus plantarum*; LB, *Lactobacillus buchneri*; PB, *Lactobacillus plantarum* + *Lactobacillus buchneri*; FM, fresh material; SEM, standard error of mean; T, treatments; D, ensiling and aerobic exposure days; T × D, interaction between T and D; ND, not detected; * Significant at 0.05. ** Significant at 0.01. Different capital letters indicate significant differences among different treatments on the same ensiling and aerobic exposure days (*p* < 0.05); different lowercase letters indicate significant differences among different ensiling and aerobic exposure days under the same treatment (*p* < 0.05); no or same letter indicate not significant (*p* > 0.05).

**Table 5 microorganisms-11-00513-t005:** Effects of additives and days of anaerobic fermentation and aerobic exposure on microbial alpha diversity of native grass silages.

Items	Treatment	Days						Significance			
		2	7	15	60	B4	B8	SEM	T	D	T × D
OTU	CK	21.67Ac	76Aa	57.33Aab	39.67Abc	37.67ABbc	23.00Bc	2.34	**	**	NS
	LP	18.33Ab	17Bb	16.33Bb	54.33Aa	53.67Aa	63.33Aa				
	LB	26.00Ab	20.67Bb	17.33Bb	41.00Aab	38.67ABab	51.67Aa				
	PB	19.00Aab	12.00Bb	16.67Bab	36.67Aa	25.00Bab	29.33Bab				
Chao1	CK	36.44Ab	101.33Aa	70.00Aab	52.76Ab	48.70Ab	30.71Bb	3.34	NS	*	**
	LP	26.11Ab	20.33Bb	33.00Ab	86.60Aa	68.82Aa	76.83Aa				
	LB	42.23Aa	30.67Ba	45.50Aa	74.00Aa	63.11Aa	62.76Aa				
	PB	28.90Aab	18.42Bb	29.11Aab	51.46Aab	42.73Aab	54.29ABa				
Shannon	CK	0.90Ab	3.07Aa	1.14Ab	1.31Ab	1.48Ab	0.85Bb	0.14	**	**	**
	LP	0.33Ac	0.55Bbc	0.29Bc	1.93Ab	0.97ABbc	3.37Aa				
	LB	1.34Aa	0.54Bbc	0.24Bc	0.78Ab	0.79BCb	1.06Bab				
	PB	0.63Aab	1.01Ba	0.89Aab	0.93Aab	0.32Cab	0.29Bb				
Simpson	CK	0.29ABb	0.77Aa	0.27Ab	0.36Ab	0.48Aab	0.27Bb	0.03	*	**	**
	LP	0.10Bc	0.17Cc	0.07Bc	0.54Aab	0.25Bbc	0.77Aa				
	LB	0.56Aa	0.15Cbc	0.06Bc	0.24Ab	0.24Bb	0.27Bb				
	PB	0.22ABbc	0.48Ba	0.38Aab	0.22Abc	0.09Bc	0.06Bc				
ACE	CK	79.40Aa	96.98Aa	74.76Aa	67.29Aa	67.62Aa	40.45Ba	4.39	NS	*	NS
	LP	32.83Ab	24.64Bb	42.03Ab	94.75Aa	89.28Aa	93.54Aa				
	LB	53.05Aab	43.60Bb	44.07Ab	122.99Aa	77.94Aab	72.6ABab				
	PB	34.58Aa	31.91Ba	45.79Aa	66.23Aa	53.19Aa	75.22ABa				
Coverage	CK	0.99	0.99	0.99	0.99	0.99	0.99	0.00	NS	NS	NS
	LP	0.99	0.99	0.99	0.99	0.99	0.99				
	LB	0.99	0.99	0.99	0.99	0.99	0.99				
	PB	0.99	0.99	0.99	0.99	0.99	0.99				

CK, control group; LP, *Lactobacillus plantarum*; LB, *Lactobacillus buchneri*; PB, *Lactobacillus plantarum* + *Lactobacillus buchneri*; FM, fresh material; SEM, standard error of mean; T, treatments; D, ensiling and aerobic exposure days; T × D, interaction between T and D; NS, not significant; * Significant at 0.05. ** Significant at 0.01. Different capital letters indicate significant differences among different treatments on the same ensiling and aerobic exposure days (*p* < 0.05); different lowercase letters indicate significant differences among different ensiling and aerobic exposure days under the same treatment (*p* < 0.05); no or same letter indicate no significant (*p* > 0.05).

## Data Availability

Data is contained within the article.

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
