# Peer review of "Effects of Different Types of LAB on Dynamic Fermentation Quality and Microbial Community of Native Grass Silage during Anaerobic Fermentation and Aerobic Exposure"

_microorganisms, 2023, doi:10.3390/microorganisms11020513_

Round 1
Reviewer 1 Report
I have revised the manuscript, and here are the comments:-
The authors need to check the language of the manuscript.
Many abbreviations in the abstract need to be cleared, and add more results in the abstract.
Provide the origin and model of all devices.
In addition, many scientific names in the references must be written in italic format.
To improve the quality of the paper, update the reference list by adding 2022 and 2023 references.
Please follow the authors' instructions on how they write the reference in the list. Why are you capitalizing the first letter of every word? Please see the journal style. For references about textbooks, please add the page numbers of the textbook. Also, please add the city of the publisher; please see references 67 and 69.
Please make all tables self-explanatory and do not use abbreviations in the table footnote or the table legend.
Enhance the resolution of figure 2.
The conclusion shouldn’t contain results, check and rewrite this part.
Check language by an expert in the whole manuscript.
Reviewer 2 Report
- Manuscript under the title: "Effects of Different Types of LAB on Dynamic Fermentation Quality and Microbial Community of Native Grass Silage During Anaerobic Fermentation and Aerobic Exposure" is well designed and conducted. Few comments that can be improved as following:
- Writing method is high difficult with lots of abbreviation that may be confusing to the reader.
- In results: Lots of data were represented the tables that are very difficult to be understood. These data can be summarized in a graph forms to be easily understood.
Reviewer 3 Report
The article by Zhang and coworkers reports that the addition of different lactic acid bacteria could further the native grass silage quality by changing the bacterial community structure. The manuscript is well written and easy to follow, and the experiments appear to be conducted according to the standards of the field. This work is appropriate for publication in Microorganisms, and will be of great interest to the readership of this journal.
Here are some minor comments:
Why Lactobacillus plantarum (LP) and Lactobacillus buchneri (LB) are chosen for this research? As both of them have been investigated respectively.
In addition to LP and LB, do other lactic acid bacteria affect the quality and microbial community of native grass silage?
Round 2
Reviewer 1 Report
The authors have carefully processed all comments. The quality of the manuscript has increased significantly. I have no further comments.